# Determinants of Children’s Fruit Intake in Teso South Sub-County, Kenya—A Multi-Phase Mixed Methods Study among Households with Children 0–8 Years of Age

**DOI:** 10.3390/nu13072417

**Published:** 2021-07-14

**Authors:** Eleonore Kretz, Irmgard Jordan, Annet Itaru, Maria Gracia Glas, Sahrah Fischer, Thomas Pircher, Thomas Hilger, Lydiah Maruti Waswa

**Affiliations:** 1Center for International Development and Environmental Research (ZEU), Justus Liebig University Giessen, 35390 Giessen, Germany; Eleonore.Kretz@ernaehrung.uni-giessen.de (E.K.); Maria.G.Glas@zeu.uni-giessen.de (M.G.G.); 2School of Public Health and Biomedical Science and Technology, Masinde Muliro University of Science and Technology, Kakamega P.O. Box 190-50100, Kenya; itaruannet@gmail.com; 3Institute of Agricultural Sciences in the Tropics (Hans-Ruthenberg-Institute), University of Hohenheim, 70593 Stuttgart, Germany; Sahrah.Fischer@uni-hohenheim.de (S.F.); Thomas.Hilger@uni-hohenheim.de (T.H.); 4Research Center for Global Food Security and Ecosystems (GFE), University of Hohenheim, 70593 Stuttgart, Germany; Thomas.Pircher@uni-hohenheim.de; 5Department of Human Nutrition, Egerton University, Egerton P.O. Box 536-20115, Kenya; lwaswa@egerton.ac.ke

**Keywords:** children, fruit consumption, fruit availability, TIPs, fruit trees, nutrition education, Kenya

## Abstract

Fruits are micronutrient-rich sources which are often underrepresented in children’s diets. More insights into the determinants of children’s fruit consumption are needed to improve nutrition education in Teso South Sub-County, Kenya. A multiphase mixed method study was applied among 48 farm households with children 0–8 years of age. A market survey together with focus group discussions were used to design a formative research approach including qualitative and quantitative data collection methods. The unavailability of fruits and the inability to plant fruit trees in the homesteads were the main challenges to improve fruit consumption behaviour, although a number of different fruit species were available on the market or in households. Perceived shortage of fruits, financial constraints to purchase fruits and taste were important barriers. Fruits as snacks given between meals was perceived as helpful to satisfy children. The mean number of fruit trees in the homesteads was positively associated with fruit consumption. Field trials are needed to test how best fruit trees within home gardens and on farms can be included, acknowledging limited space and constraints of households with young children. This should be combined with nutrition education programs addressing perceptions about the social and nutrient value of fruits for children.

## 1. Introduction

In recent years, Kenya has seen declining rates of malnutrition such as stunting, wasting and underweight, but the population is still affected by poor nutritional status [1]. The national stunting rate is considered high and a matter of public health concern (26% of children 6–59 months of age are stunted) [2]. At the same time the double burden of malnutrition, coexistence of undernutrition and overweight and obesity are present not only in urban but also in rural areas [3,4,5].

The main determinants for stunting are presumed to be found during the rapid development of a child in utero and during the first 2 years, which is also referred to as “the first 1000 days”. This period, the so called “window of opportunity” is a critical period for optimal growth as well as for the health and behavioural development of a child [6,7]. Among others, the causes for stunting are intrauterine growth retardation, inadequate nutrition of infants and young children with poor breastfeeding and complementary feeding practices, as well as recurring infections during early life [7]. An adequate nutrient supply during this sensitive time is essential to prevent the effects of chronic malnutrition. However, children above two years of age may be more sensitive to interventions and should therefore not be neglected [8].

Besides the provision of macronutrients, several micronutrients are thought to be critical for the growth in early and later childhood such as vitamin A, D and K, as well as zinc, iron and iodine [9]. The Kenya National Micronutrient Survey in 2011 stated that iron deficiency was prevalent in 21.8% of children between the age of 6–59 months with a peak of 34.6% in children 12–23 months of age, indicating complementary foods to be low in bioavailable iron [10]. The prevalence of vitamin A deficiency in pre-school children was classified as mild (9.2%) by the World Health Organisation (WHO), whereas zinc deficiencies among this age-group posed a major public health issue with a prevalence of 83.3% [10]. These findings are of no surprise considering that complementary foods in Kenya are mainly cereal based containing high contents of phytates, which contribute to a low intestinal absorption of zinc and iron in young children [11,12].

Fruits, as well as vegetables and animal source foods, are micronutrient rich sources which are usually underrepresented in children’s diets in Western Kenya [13,14]. The nutritional value of fruits arises from their contents in vitamins, carotenoids, minerals, fibre and antioxidants that they contain [15,16,17]. Further benefits of fruits include the potential to improve the bioavailability of plant-based iron and other minerals due to their high levels of vitamin C [18]. Since fruit and vegetables are usually combined in listings of the dietary diversity score for complementary foods [19], limited data is available on fruit consumption alone in young children. For Kenyan adults fruit consumption has been determined within a cross-sectional study in 2015 by Pengpid and Peltzer [20], showing that on average 0.78 servings of fruits and 1.31 servings of vegetables were consumed in a day, not meeting the recommended five servings (in total 400g) of fruits and vegetables per day [21]. Although fruit consumption was generally higher in children than in adults in Western Kenya, a daily intake was only achieved by 19% or 30% of children, depending on the season [13,16,22]. The main reasons for this low fruit consumption were described by Kehlenbeck et al. [23] as relating to a lack of consumer awareness, consumer preference, degradation of natural vegetation, lack of tree domestication techniques, lack of processing facilities, and poor marketing pathways.

In order to identify feasible and acceptable recommendations for nutrition education interventions which help to increase (young) children’s fruit intake, it is necessary to gain more insight into the determinants of children’s fruit consumption. Therefore, this study aimed to identify factors which facilitate and hinder the improvement of fruit consumption among children 0–8 years of age in Teso South Sub-County in Kenya.

## 2. Materials and Methods

This study was implemented as part of the research within the Education and training for sustainable agriculture and nutrition in East-Africa (EaTSANE) project. The international research project brings together multidisciplinary stakeholders to make farming practices more sustainable and improve diets of households in Kenya and Uganda. It uses a participatory action learning approach to diversify the food system [24]. The design of the EaTSANE project (2018–2021) was informed by findings from a former project called HealthyLAND (2015–2019). It is part of the LEAP-Agri Program, an initiative of the African Union and European Union leveraging agriculture and nutrition research [25].

In this study, we applied a multiphase mixed method study design. This combined focus group discussions and a market survey along with agriculture activities on a demonstration plot, with a formative research approach over different project phases [26]. The demonstration plots were established close by the homesteads of the study participants. The data collected in the market survey and focus group discussions were used to design the formative research and quantitative data collection (Figure 1). The data from the formative research (Trials of Improved Practices (TIPs)) and the quantitative data collection were finally triangulated.

### 2.1. Study Site and Participants

This study was conducted in Teso South Sub-County in Western Kenya, which borders Uganda. The region experiences usually two rain and two dry seasons per year which influences agricultural production. Due to climate change the seasons in the study area are becoming quite unpredictable [27]. During the time of data collection, we faced periods of sufficient rain fall for fruit planting during the TIPs process. The markets of Teso South, close to the Ugandan border, close to Mt. Elgon, and close to one of the major trade road routes, also feature fruits that are off season in Teso South itself. Therefore, fruits available in Teso South do not necessarily need to be in season. This pertains particularly to fruit production originating in Uganda since the prices for fruits are higher in Kenya, and therefore is the favoured region of selling for Ugandan producers (see also Table 2).

Eight villages were purposively selected for this study, based on their close proximity to agriculture-based demonstration and training plots established as part of agricultural research activities in the EaTSANE project and having been included by the previous HealthyLAND project. Within these villages, 53 families were purposively selected from a cohort established in the former research project which had been implemented by the principal investigators in the same region between 2015–2018. At the cohort’s baseline survey, we had included farm families with children below 5 years of age. Hence, when this study was conducted the oldest children of this cohort were 8 years old. We also included families with children up to the age of 8 years as they are still likely to depend on what is served at household level. The eligibility criteria to participate in this study were thus being a smallholder farmer with a child below the age of 9 years living in one of the selected villages close to the demonstration sites of the EaTSANE project. Participation in the study was on a voluntary basis and informed written consent was obtained from the participants prior to data collection.

The sample size was estimated based on the core activity in this study, the TIPs. The Manoff Group who established TIPs in the field of social behaviour change communication suggested a sample size of 20–50 households [28]. In this study we aimed at 50 rural farm households with a focus on dietary diversity.

### 2.2. Demonstration Plots

The agricultural demonstration plots were set up in a randomised complete block design to test two cropping systems which were evaluated together with households from the study area. These systems were (i) the three-sister-system (intercropping of maize, climbing beans and pumpkin) and (ii) the Three Strata Food System. The system was adapted from the Three Strata Forage System [29], and attempts to increase agrobiodiversity in crop production, without causing competition between different species, and producing biomass to increase soil fertility. The system is rather flexible whereby early fruiting species such as papaw (*Carica papaya*) or passion fruits (*Passiflora edulis*) were integrated in the third stratum together with species that start fruiting later. Hence, fruits become already available in the second year after their establishment. Additionally, fruit species such as or physalis (*Physalis peruviana)* can be grown in the second stratum.

The Three Strata Food System used in this project focussed mainly on integrating nutrient-dense crops into different vertical and horizontal levels, by using (fruit) trees and hedges as border plants. For this, a plot of land was divided into a core area for annual crops surrounded by three strata of multifunctional legumes (potentially providing food, fodder, soil improvement, and other ecological functions) hedges, and trees (Figure 2). The participating households had been introduced to the demonstration plots and the objectives during a workshop in July 2019.

### 2.3. Seasonal Fruit Availability

In July 2019 a market survey on fruit availability was conducted within the study area. A list of markets frequented by the participants in the study was compiled by the research assistants. These markets were visited to list the fruits on offer, together with prices, their origin and the type of storage at the food stall. The point of saturation in terms of information about availability and prices was reached after visiting five markets; no further markets were visited.

The information from the market survey was used as basis for 12 focus group discussions which focused on the perceived availability and value of fruits and vegetables in the upcoming season. The discussions were held with participating households, men and women separately, and young people from four villages. Both activities were conducted to inform the following data collection phase and the design of the list of recommendations to be tested in the following formative research: Trials of Improved Practices.

### 2.4. Trials of Improved Practices (TIPs)

The Trials of Improved Practices (TIPs) were conducted between August and October 2019, following up on recommendations of the Manoff Group who had established the formative research technique in the field of social and behavioural change [28]. In this study we aimed at testing the feasibility of improved child feeding practices by enhancing fruit intake among the sampled households, and at identifying factors facilitating and hindering their implementation. The primary caregivers of the children were offered a choice of recommendations to try out, as well as the opportunity to respond and give reasons for their choices in a consultative process. Using these methods, constraints and motivators for a behaviour change towards better health and nutrition were investigated [31].

Four trained enumerators facilitated the conversations at household level in the local language Teso or Kiswahili. The training of the enumerators included topics like “principles for implementing TIPs”, “how to counsel participants on recommended child feeding practices” and “how to respond to possible upcoming difficulties in their realisation”, as well as “how to fill in the monitoring sheets”. Refresher training was given regularly on difficulties encountered in the process and how to negotiate with the caregivers about the various options offered during TIPs to improve child feeding practices.

A total of three visits were conducted in each household (Figure 3). During the first TIPs visits the facilitators interviewed the primary caregiver, most often the mother, about the actual child feeding practices within that particular household. The facilitators gathered information by asking the caregivers about their daily child feeding practices and noted down the answers. Based on the feedback from the respondents, the facilitators reviewed them and selected the practices that needed improvement. They counselled the caregivers on the issues that needed to be improved by providing appropriate recommendations and the benefits to expect. Therefore, not all caregivers received the same key recommendations, but a suitable selection according to their current practices. The interview guides that were used for the first visits included questions about possible barriers for the improved practices, the respondents’ reactions to the proposed practices and the recommendations that were finally agreed to test. Caregivers who were already following the recommended practices were encouraged to continue doing so and also challenged to try out more improvements to enhance their children’s diets.

Two to three weeks after visit one and two respectively, assessment and counselling proceeded during the follow-up visits according to additional or arising problems. The interview guides for the second and third visits included household-specific questions about the experiences of the participants with the trial and the intention to continue with the improved practices in the future.

### 2.5. Key Recommendations Tested during the Trials of Improved Practices (TIPs)

Table 1 shows a catalogue including 13 key recommendations for child feeding practices which were used in the TIPs and were related to five topics: (a) fruit consumption and availability, (b) nutritious meals and snacks, (c) porridge quality, (d) responsive feeding, and (e) priorities during mealtimes. Two recommendations specifically addressed fruit consumption and availability: (a) serve the child fruit on a daily basis and (b) plant more fruit trees in the homestead. The latter recommendation, the planting of fruit trees in the respondents’ homesteads and fields, was linked to the activities on the agricultural demonstration and training plots of EaTSANE where fruit trees had been included into the Three Strata Food System.

### 2.6. Fruit Trees Assessment

During the second household visit, the TIPs facilitators assessed the number of different fruit tree species in the homesteads of each household. For this, respondents were asked what different types of fruit trees they had in their homesteads or farms, which was verified by the facilitators’ observations, and all responses were listed.

### 2.7. Data Analysis

The market data and focus groups’ discussion results were summarised and used to inform the trainers and TIPs facilitators on access to and (perceived) fruit availability in the upcoming season in the study region. The findings were also used to design recommendations to be tested during the TIPs.

Implementation rates were calculated, representing the percentage of caregivers who successfully implemented the recommendations on child feeding practices out of the number of caregivers who agreed to try the practice. 

The data on barriers and motivators to implement improved practices collected during the TIPs visits were analysed by applying a structuring qualitative content analysis according to Mayring [32]. The data were primarily categorised following the TIPs thematic structure. Subcategories were inductively and gradually derived from the respondents’ answers. The categories were defined and a coding guideline developed. The coding process was done using the open-source software “QDA Minor Lite v1.4.1” [33]. The coding was repeated independently by a second coder for consensual validation of the coding process [26]. During the process, disagreements were discussed amongst the coders and definitions for specific codes were clarified and the coding guideline revised. As there are different coefficients used in qualitative research to calculate intercoder agreement with no consensus for one specific, Cohen’s Kappa, Brennan and Prediger, Scott’s Pi, Gwet’s AC and Krippendorff’s Alpha were computed after the final coding process of the data. All results showed high intercoder agreement with a coefficient of at least 0.91 ± 0.01 (*SE*); (*p* < 0.001).

### 2.8. Triangulation

The quantitative analysis of how many caregivers carried out inadequate practices and how many successfully tried out the suggested practices was performed using Microsoft Excel 2016. A qualitative cluster analysis of the TIPs findings on fruit consumption was conducted by grouping the respondents based on successfully implemented recommendations regarding child feeding practices. The findings were triangulated with the number of trees in the respondent’s homesteads [34]. Subsequently, the number of fruit tree species was tested for significant statistical differences and unstandardised effect sizes between these cluster groups using univariate ANOVA (Analysis of Variance). In case of statistical significance between the groups, a post hoc test (Tukey’s test) was performed for pair-wise comparisons between individual groups. Statistical analysis was conducted using the Software IBM SPSS Statistics Version 27.

## 3. Results

### 3.1. General Characteristics of Study Participants

The women’s ages ranged from 22 to 58 years with an average of 36 ± 10 years. They took care of one to four children below the age of 9 years. At the beginning of the trials, 53 households were included into the study. Five households dropped out during the TIPs phase. Two households moved away, and two households withdrew their approval to participate in the study due to lack of interest. One household had to be excluded from the data analysis as a result of changes in the family situation and the need for extra counselling by the facilitator, which led to biased data. Hence, 48 households were included in the final data analysis (Figure 3).

### 3.2. Key Findings from the Market Survey and Focus Group Discussions

Table 2 shows the types of fruits offered on the markets usually accessed by the participating households in July 2019. Out of the 13 different fruits found on the markets, nine were sourced from Uganda, three from Kenya itself and one (i.e., apples) from South Africa, the smaller markets having mainly sourced the fruits from the Busia market. One market had only three different fruits on offer, namely lemons, oranges and mangoes. Lemons, oranges and mangoes were all imported from Uganda and were the only fruits available on all markets, with the prices ranging from 3–10 Kenyan Shilling (KSh) per piece in the case of lemons and oranges, and 10–20 KSh per mango. Depending on the market, one could also find watermelons and bananas. Other nutritious fruits like papaya, plantains or pineapple were only found in Busia, an urban market. Avocados were available on two out of the five surveyed markets and were the only fruits which were sourced locally by the traders. 

The focus group discussions in July 2019 revealed that the households anticipated having access to between 13 and 32 fruit species in the forthcoming period July to December 2019. Out of these, very few were listed as only available in the homesteads. The awareness about the availability of fruits differed slightly between men and women, the latter often knowing more different species than men. They potentially had more knowledge of what might be available around the homestead, but less knowledge of what might be available on the markets. Participants most often mentioned having access to fruits either at the market and at home, or at the market only. Fruits which are only available within or around their homesteads were mentioned less often. The difference in the number of fruits being available at village level was a result of participants naming different varieties of e.g., passion fruit or papayas. However, all focus group discussions revealed that households would have access to a minimum of 10 different fruits which would be available either at home or on the market in the period covered by the TIPs, i.e., August–December. However, the focus group discussions revealed that the willingness of households to purchase fruits on a market was generally low.

### 3.3. Observations around the Demonstration Plots

Farmers in South-Teso practised inter-cropping as a conventional system of cropping. Inter-cropping of maize and beans was most common, as these were the main staple crops in the region. Introducing inter-cropping in the farming system in South Teso therefore blended well with the local farming system. In many households, fruit crops were likely to be found around the homestead. This could be related to the ease of establishment and maintenance within the homesteads (watering, protection from animals, and generally higher fertility of the soils) as opposed to in the field. Another reason for planting fruits close by the homestead is that it is easier to keep outsiders from harvesting the ripe fruits than it is in the fields away from the homestead. It was initially proposed to the farmers that passion fruits should be included into the trials in the fields, but farmers argued that they may not benefit much from the yield, as the ripe fruits would attract “theft” by children and youths. In fact, the few farmers who received unused fruit seedlings after the experimental planting was completed, planted all material within their homesteads. Although farmers with fruit crops could market the fruits and earn extra household income, the length of time it takes for the farmers to see the benefits was mentioned as another reason for the low interest in fruit crops.

### 3.4. Implementation Rate of Nutrition Related Recommendations

Overall, most of the key recommendations for child feeding practices were successfully implemented with high implementation rates of above 90%. Out of the 13 recommendations, the two recommendations (1) to serve fruits on a daily basis and (2) plant (a) fruit tree (s), had the lowest implementation rates, indicating that these recommendations were the most challenging for the families. In the case of the recommendation to serve fruits on a daily basis, only 50% of the 16 families who tested this recommendation managed to do so and only 14% of the 22 families who agreed to plant a fruit tree managed to do so successfully (Figure 4).

### 3.5. Seasonality as a Hindering and Facilitating Factor for Consumption

During the TIPs sessions, fruits from the list collected during the focus group discussions were recommended to be consumed as snacks or to be used to enrich children’s porridge or other meals. However, although fruits were available, “seasonality” of fruits hampered the consumption of fruits by children on a daily basis. Caregivers reported that *“fruits are not in season”* (woman, 30 yr (with child, 3 yr)) and therefore *“not available at the moment”* (woman, 25 yr (with child, 5 yr)). They reiterated their willingness to try out the recommended improved practices once the fruits were in season:

*“[She is] willing to try when fruits are in season because fruits will be readily available”* (woman, 28 yr (with child, 5 yr)).

Fruits were considered to be available when they could be harvested from trees planted within the homesteads or collected from the natural vegetation in the surrounding areas. This challenged especially those families who did not have access to a variety of fruit trees within their homesteads to cover the seasonal gaps:

*“[She] has only two [different types of] fruit trees”* (woman, 53 yr (with child, 8 yr)).

Seasonal fluctuations affected the year-round availability of certain fruits at the markets. In the course of the TIPs sessions there was also a period when no mangoes were available in the area, although the focus group discussions had revealed that mangoes would at least be available on the markets.

Regardless of whether or not the fruits were available on the markets, some caregivers reported that they were inaccessible since financial resources were needed to access them in times of fruit scarcity at the homestead:

*“When you don’t have money at the time when fruits are out of season it becomes difficult to access them”* (woman, 25 yr (with child, 5 yr)).

In some instances, the caregivers could not afford fruits, whereas in others they were able to purchase fruits at the markets:

*“[She] was able to purchase bananas, mangoes, avocado”* (woman, 30 yr (with child, 5 yr)).

The quote also confirms that fruits were actually “available” for some caregivers on the markets during the period the TIPs was conducted. This indicates that availability was linked with accessibility for those other caregivers who did not have the financial resources to access the market to purchase the fruits, or who were not willing to spend their financial resources on fruits.

### 3.6. Fruit Trees on the Homesteads

In order to reduce the effects of seasonality of fruits the TIPs facilitators discussed with the caregivers the existing fruit trees available within their homesteads. Table 3 provides an overview about the number of different types of fruit trees that were found in the homesteads. The mean number of fruit tree species available within the participants’ homesteads ranged from 1 to 10 with a mean of 4.6 (±2.5). The most common fruit species were mango (*Mangifera indica*), avocado (*Persea americana*) and jackfruit (*Artocarpus heterophyllus*), followed by dessert banana (*Musa acuminata*), papaya (*Carica papaya*) and guava (*Psidium guajava*). With the exception of bananas, these are highly seasonal fruits. 

As a long-term investment, the TIPs facilitators recommended that respondents plant a wider variety of fruit trees in order to be able to ensure year-round availability of fruits. Respondents were willing to try to plant more fruit tree species, with 23% of them highly motivated, believing they could succeed because they had planted fruit trees before and had the necessary knowledge.

However, the implementation proved to be difficult. The majority of the respondents who were not able to plant fruit trees (58%) mentioned lack of access to seeds or seedlings as reason for not being successful. The three respondents who successfully planted fruit trees obviously had access to seedlings; one reporting receiving them from neighbours. However, other difficulties in successfully planting fruit trees were reported, with seedlings and young trees being destroyed by animals or children.

*“The children uprooted the planted pawpaw seedling”* (woman, 36 yr (with child, 5 yr)).

Other difficulties which prevented respondents from planting were related to unfavourable weather conditions, in particular unreliable rainfall, and the fear of pests:

*“The pests are still a major problem that even the pesticides are not helping at all”* (woman, 25 yr (with child, 5 yr)).

### 3.7. Children’s Preferences, Lifestyle and Daily Routines, Knowledge and Perceptions

Children’s preferences influenced their caregivers’ food choices. One mother stated that her *“children like biscuits”* (woman, 41 yr (with child, 3 yr)). In consequence, caregivers had taken their children’s likes and dislikes into consideration and so provided them with less healthy options such as biscuits or other sweets. However, the preferences of family members for certain foods also functioned as motivators for providing healthier options. One mother was inspired by her children’s preferences for fruits:

*“[She is] interested as she says her children love to eat fruits and [she] can make more effort to provide it more often”* (woman, 53 yr (with child, 8 yr)).

Lack of time was a difficulty that was experienced by caregivers during the TIPs which was affecting food availability at household level. For example, time constraints made it difficult for the caregiver to go to the market to purchase fruits:

*“[I] rarely get time to go look for fruits at the market”* (woman, 42 yr (with child, 2 yr))

Time constraints were also linked to periods of sickness that the respondents experienced during the trials. One caregiver did not have time to plant fruit trees due to her hospital admission; others also reported about sickness:

*“I forgot to plant! I was sick and unable to attend to farm work”* (woman, 46 yr (with child, 8 yr))

Lack of knowledge about the importance of fruits in the diet has influenced the consumption level and provision of fruits to the children:

*“Had never thought that it is important to have fruits throughout the year”* (woman, 36 yr (with child, 5 yr))

Another factor needed to improve fruit consumption was knowledge about the benefits of fruits for their children’s health. One mother stated as a motivational factor to plant more fruit trees:

*“Fruits are good for our health”* (woman, 46 yr (with child, 5 yr))

Serving fruits as snacks or extra meals needs extra effort and might put an additional burden on the mother/caregiver. To minimise the burden one recommendation was to use fruits to enrich (children’s) porridge. This practice was seen differently by the caregivers. Some mothers perceived porridge enriched for example with lemon juice as being tasty, while others perceived the practice as being unnecessary for their children:

*“[I] think the child is grown enough [so] that she does not need more enriching”* (woman, 34 yr (with child, 8 yr))

At the same time mothers reported their own observations and positive changes in their children’s behaviour. An indirect positive effect of fruits which were served as snacks could be seen in statements about the regular provision of snacks to children:

*“[…] helped the children not to be restless before meals, they are more satisfied and do not disturb [me while I] […] prepare the meals”* (woman, 47 yr (with child, 6 yr)).

### 3.8. Implementation Patterns and Fruit Trees in the Homestead per Cluster Group

The cluster identification on successful implementation of child feeding recommendations resulted in four groups: the barrierless, the overcomers, the limited in fruit consumption and/or planting fruit trees, and the struggling (Table 4). The first two groups, the barrierless and the overcomers, managed to implement all recommendations which were agreed upon during the trials. The other two groups had specific challenges with regard to the implementation of the recommendations in regard to fruit consumption and the planting of fruit trees.

The mean number of fruit trees in the homesteads differed significantly between the cluster groups (*p* = 0.000; *α*-Level: 0.05, univariate ANOVA). The following Tukey’s test displayed that the mean number of fruit tree numbers of the cluster groups 1 and 2 did not differ significantly from each other. The same applied to groups 3 and 4 (Table 5).

## 4. Discussion

In this study, the availability of fruits at the homestead was a key driver to include fruits into the children’s diets and mother/caregivers explained that it would be even easier for them once the fruits were in season. These findings do not necessarily imply that fruits were not available at the markets and so should, perhaps, rather be described as “perceived unavailable”. The challenges of fruits being unavailable and the resulting high costs especially in off-seasons were also reported by caregivers of 6–23 months old children in Migori, Kenya [14]. East African farms often feature wild, indigenous (semi-domesticated) and domesticated fruit trees, either on the farm or in its vicinity. In Eastern Kenya, over half of the fruit trees found on farms were found to be indigenous fruit trees [35]. A wide range of other wild and indigenous fruit trees is also found off-farm [23]; however, this is not reflected in the consumption patterns. Cultivating a variety of “priority species” covering the different seasons could help to improve the fruit supply throughout the year which is independent of financial constraints of the households. The positive association between a more successful implementation of the recommended child feeding practices and a high number of different fruit trees at the homestead found in this study confirms the value of these attempts.

To achieve variety in fruit cultivation, indigenous fruits in particular could play an important role as they are easier to grow, more resilient to pests and are a low-cost source of micronutrients [36]. The study from Ekesa et al. highlighted a low consumption and access to indigenous fruits in a rural area in Matungu Division in Western Kenya, and suggested promoting their cultivation [36]. Further benefits of fruit trees are their wide and deep root systems which make them less sensitive to drought than annual staple foods such as maize. They will therefore yield a harvest even if the staple foods fail [23]. However, indigenous fruit trees are often semi-domesticated and can, therefore, vary in fruit quality and taste [23]. Additionally, indigenous fruit trees often have a low market value and may carry the stigma of being poor people’s or children’s food [23,37].

In this study, the recommendation to plant more fruit trees in the homestead turned out to be a challenge for the participants. Often fruit species need some years after planting before a first harvest can be expected. Nevertheless, there are various cultivars and species available which may be harvested already in the second year after establishment. However, seedlings need a proper handling in the home garden area before field establishment to survive dry periods and fruit tree selection towards drought resistance is important. Although the respondents were motivated by the prospect of increased fruit availability in future, they were deterred by the low availability of seedlings, unfavourable weather conditions, the destruction and uprooting of seedlings by animals or children and former experiences with pests. Other studies showed that unknown seed dormancy periods and unreliable germination, such as with the baobab tree, deterred farmers from planting fruit trees [37]. Regardless of these difficulties, it was also challenging for the researchers themselves to get access to seeds or seedlings of the fruit species needed for the demonstration plots set up within the EaTSANE project. However, Keding et al. stated in their study that only 16% of their respondents in Western Kenya mentioned access to fruit tree seedlings as a promising strategy to enhance their fruit consumption [16]. In the same study, only 4% considered increased knowledge about fruit cultivation to be a successful strategy to increase their fruit intake. The availability of seedlings may thus be a contributing but not the driving factor in low fruit consumption.

Financial constraints were mentioned by the participants as a reason not to purchase fruits on the market, which reflects the lower priority given to fruits. A study in Nairobi investigated the vegetable and fruit consumption of its urban population and showed that basic food expenditure emphasised animal source foods, which accounted for 40% of the budget while 34% was spent on staples, 18% on vegetables, and only 8% on fruits [38]. The same study also revealed a steady increase in fruit expenditure with a rise in income, i.e., the lowest-income households purchased about half of the quantity of fruits compared to the top one-fifth of consumers [38]. These results corresponded with findings from a study in a rural area in Western Kenya, showing that households spent the smallest amount of the food budget on fruits [16]. A lack of awareness of the health benefits of regular fruit consumption is suspected in the population in sub-Saharan Africa, which could contribute to the low priority given to purchasing fruit [23]. Although the respondents of the current study attributed nutritional value and positive health effects to fruits, they also named indigenous fruits as “poor man’s food”, which leads to the assumption that fruits are often perceived as having limited value.

The children’s preferences for or dislikes of certain foods directly influenced the food choices made by mothers/caregivers, and the children’s reactions to improved practices and healthier choices facilitated or hindered their implementation. Another TIPs trial conducted in Western Kenya (Kisumu and Migori) referred to the problem of mothers/caregivers offering only their preferred foods to children, because children tend to reject unknown foods. Repeatedly offering “new foods”, modifications in preparation methods and different food combinations facilitated their acceptance [14]. Consequently, if offering fruits does not have a social value, which seems to be the case for many indigenous fruits, nutrition messaging needs to create or improve the social value. This may be more important than emphasising the nutritional value and health aspects associated with the consumption of fruits.

Time was a crucial factor for the implementation of improved feeding practices. Respondents in this study mentioned their involvement in farming activities as reasons for time constraints, which also reduced their time to go to the market and purchase fruits. Women often face a triple burden of work with housework, childcare and subsistence farming activities. Heavy workloads therefore compete for time with that required for adequate childcare and feeding practices. “*Competing interests for caregivers’ time*” as a barrier to implementing improved child feeding practices in Rwanda were reported by Williams et al. [39]. Farming activities that prevented mothers from preparing meals were also reported by Muehlhoff et al. for caregivers in Cambodia and Malawi [40]. Clearly, this is not a new finding but needs to be addressed in future nutrition education campaigns by offering time saving solutions in complementary and family food preparation. 

The positive effects of serving fruits to children in between meals so that the less-hungry children could allow their mothers to finish their chores, thus creating more opportunities for women’s own care, could be emphasised in nutrition education messaging.

### Fruit Trees in the Homesteads as a Determinant for Behavioural Change in Fruit Consumption

The significant difference regarding the number of fruit trees between the groups who were able to implement all recommendations agreed upon (group 1 and 2) and the ones who did not implement all recommendations agreed upon (group 3 and 4) confirmed the important role of fruit trees for successful implementation of the improved child feeding practices. One explanation could be a higher availability of fruits, which facilitated not only the direct recommendation to serve fruits on a daily basis, but also to enrich porridge, serve as a snack, serve a nutritious breakfast or generally serve a variety of foods. However, growing fruit trees takes time and space and informal discussions held with farmers revealed that fruit trees are sometimes cut down to avoid conflict with neighbours during harvest time. This might be related to e.g., excess fruit availability during harvest time, with challenges in marketing these fruits and neighbouring children climbing the trees, with the potential health risks involved if they fall. At the same time limited access to land and land titles may deter farmers from planting perennial crops like trees. The actual number and diversity of fruit trees varied strongly between the different households. Another study in the same region found a positive correlation between fruit tree diversity and decreasing farm size, mainly attributed to the multi-purpose possibilities offered by many trees [41]. Trees can provide not only foods such as fruits, but also timber, medicines, fodder and different ecosystem services [35]. Nyaga et al. [42], found that the diversity of fruit tree species increased with decreasing resource endowment. A high diversity of fruit trees is important as the lower the diversity, the higher the seasonal gaps when no fruits are available [33,41]. Nutrition education campaigns could raise awareness of different fruit trees, as an increased tree cover has been associated with increased household fruit and vegetable consumption in another study [35]. Additionally, many wild and indigenous fruit trees (which are more affordable and accessible in many areas than high-value domesticated cultivars) are currently underused, therefore offering a high potential for increasing fruit consumption [23].

## 5. Strengths and Limitations

The strength of this study was that the repeated visits allowed an intensive dialogue with the primary caregivers of the participating households. This enabled the research team to get detailed information about the barriers and facilitating factors for improving child feeding practices. However, the recommendations did not differentiate between wild, indigenous and domestic high value trees. Consequently, the findings on the difficulties in consumption of fruits can only be generalised to all fruits. The findings also do not allow a detailed discussion about potential cultivars and the support needed to become more successful in planting fruit trees, especially as there are, e.g., in the case of mango, high and low value cultivars, some of which are considered almost wild. Another limitation might have been that the demonstration plots were only initiated 6 months prior to the TIPs, so that the households could not yet have benefitted from the training. 

TIPs started with a group of 53 families of which 48 households were included in the data analysis. This is quite a small sample size compared to quantitative surveys and two households less than desired as two moved away and two households did not want to continue. However, the three consecutive visits summed up to 144 in-depth interviews leading to point of saturation where no new information was collected. The individual counselling involved in this process is likely to achieve higher prevalence of behaviour change compared to e.g., community-based interventions but the findings of this approach can be considered to be consistent predictors for change [28].

## 6. Conclusions

The unavailability of fruits and the inability to plant more fruit trees in the homesteads were the main obstacles to increasing fruit consumption among children in Teso-South, Kenya. We may have been too ambitious with the recommendation to plant trees without offering relevant training, acknowledging the limited space and the constraints households with small children face if the trees are not planted in a protected area. Integrated production systems supported by appropriate training may therefore help in establishing more fruit trees at homesteads to provide a better seasonal supply. Additional field trials are needed with less focus on yield of staple crops, but to test how fruit trees both within home gardens and on farms can best be included to enhance fruit availability throughout the year. A wide variety of different species of fruit trees with the promotion of indigenous fruits seems to be a promising approach for the research area. At the same time, barriers in planting fruit trees need to be addressed, in particular with facilitating access to seedlings. Grafting of fruit trees may improve the quality of the fruits and reduce the time till the first harvest, which may better integrate with nutrition education campaigns. In addition, perceptions about the social and nutrient value of fruit, in particular indigenous fruits, should be addressed in nutrition education, pointing out the health benefits to ensure that appropriate amounts of fruit are consumed by all family members.

## Figures and Tables

**Figure 1 nutrients-13-02417-f001:**
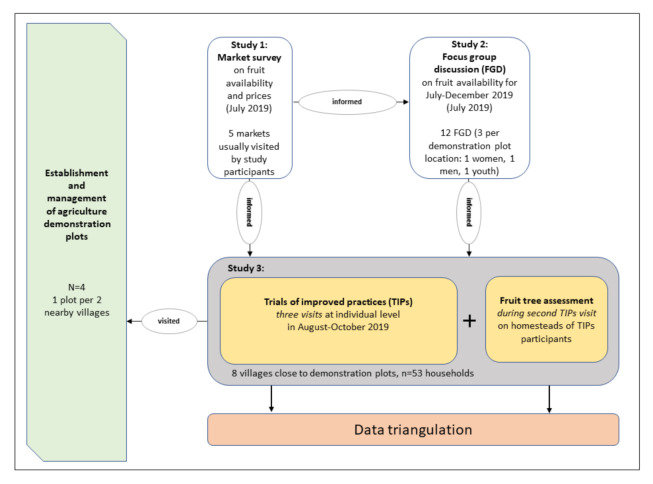
Multiphase mixed method study design.

**Figure 2 nutrients-13-02417-f002:**
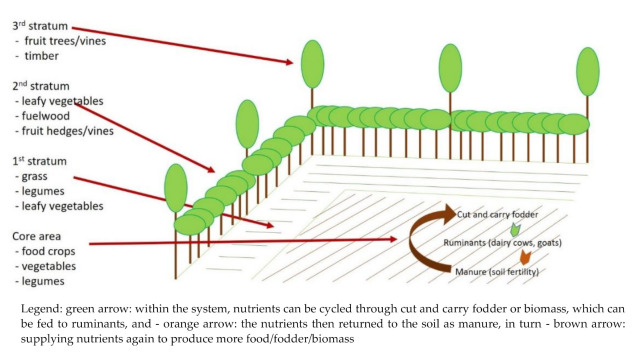
Three Strata Food System adapted from the Three Strata Forage System from Nitis et al. [30]. Image made by Thomas Hilger.

**Figure 3 nutrients-13-02417-f003:**
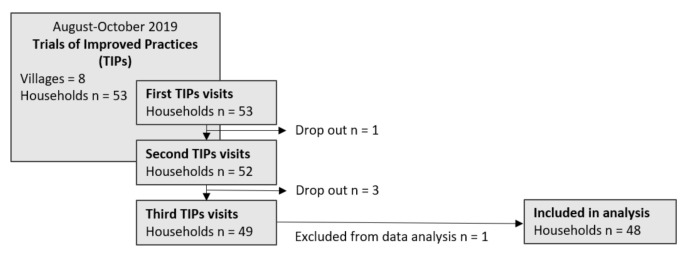
Study flow and drop-out rates of the Trials of Improved Practices (TIPs).

**Figure 4 nutrients-13-02417-f004:**
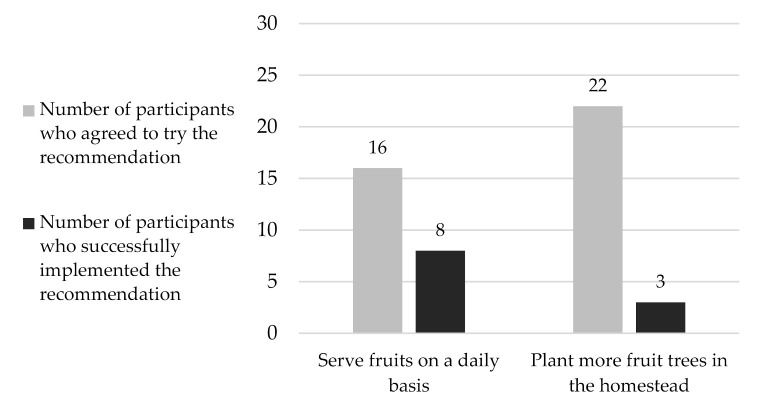
Numbers and implementation rates of the recommendations on fruit consumption tested during the TIPs (N = 48); TIPs = Trials of Improved Practices.

**Table 1 nutrients-13-02417-t001:** Key recommendations on child feeding practices tested during the Trials of Improved Practices (TIPs).

Topic	Key Recommendation Delivered During TIPs
Fruit consumption and availability	Serve the child fruits on a daily basis.
	Plant more fruit trees in the homestead.
Nutritious meals and snacks	Serve the child a nutritious breakfast (e.g., enriched porridge).
	Serve the child the actual family food instead of serving the broth of foods.
	Serve the child alternatives instead of black tea.
	Serve the child a variety of foods on a daily basis by including foods from different food groups.
	Serve the child nutritious snacks in between meals.
Porridge qualities	Prepare a porridge of a thick consistency.
	Enrich the porridge with other foods.
	Add dried vegetables (e.g., dried green leafy vegetables) to the porridge.
Responsive feeding and priorities during mealtimes	Make child feeding a priority in your household. Serve young children first.
	Make sure they get and eat their share.
	Separate the child’s bowl from the mother’s in order to know how much the child has eaten; mind the recommended amount and frequency.
	Interact with the child during mealtimes and actively and lovingly encourage him/her to eat; do not force or threaten your child to eat.

**Table 2 nutrients-13-02417-t002:** Fruits, price per piece and origin available at markets accessed by study participants end of July 2019.

	Busia (Town)	KSh #	Origin	Lukolis	KSh #	Origin	Adungosi	KSh #	Origin	Angorom	KSh #	Origin	Asing’e	KSh #	Origin
1.	Lemon	6	Ug	Lemon	5	Ug	Lemon	3	BSm	Lemon	5	BSm	Lemon	4	Ug
2.	Orange	8	Ug	Orange	5	Ug	Orange	5	BSm	Orange	10	BSm	Orange	5	BSm
3.	Mango	15	Ug	Mango	18	Ug	Mango	10	BSm	Mango	10	BSm	Mango	10	Ug
4.	Pineapple	100	Ug				Pineapple	na	na	Pineapple	50	BSm			
5.	Watermelon	300	Ug				Watermelon	na	na	Watermelon §	10	BSm			
6.	Banana	10	Ug	Banana	6	Ug				Banana	10	BSm			
7.	Avocado	15	Busia	Avocado	20	Teso-South									
8.	Coconut	80	Ky												
9.	Thorn melon	40	Ky												
10.	Tamarind *	10	Ug/Ky												
11.	Passion fruit	3	Ug												
12.	Papaya	135	Ug												
13.	Apple	30	SA												

# Rounded average price per piece * Per handful, § per slice, Ug: Uganda, Ky: Kenya, SA: South Africa, BSm: Busia, Sofia market, na: not assessed.

**Table 3 nutrients-13-02417-t003:** Overview of the number of different types of fruit trees in the homesteads.

			TIPs Participants *n* = 47 (1 Household = Missing Value)	
	Types of Different Fruit Trees in the Homesteads		*n*	%
1	Mango	*Mangifera indica* L.	36	77
2	Avocado	*Persea americana* Mill.	36	77
3	Jackfruit	*Artocarpus heterophyllus* Lam.	31	66
4	Banana	*Musa* x *paradisiaca* L.	29	62
5	Papaya	*Carica papaya* L.	20	43
6	Guava	*Psidium guajava* L.	19	40
7	Tamarind	*Tamarindus indica* L.	9	19
8	Jamun fruit ^1^	*Syzygium cumini* L.	8	17
9	Orange	*Citrus sinensis* L.	8	17
10	Lemon	*Citrus limon* L. Osbeck	7	15
11	Passion fruit	*Passiflora edulis* Sims	6	13
12	White sapote ^2^	*Casimiroa edulis* La Llave	3	6
13	Sparse lote	*Diospyros lotus* L.	2	4
14	Pineapple	*Ananas comosus* (L.) Merr.	2	4
15	Thorn melon	*Cucumis metuliferus* E. Mey	1	2
16	Palm fruit	*Borassus flabelliformis* L.	1	2
17	Wild grapes	*Vitis vinifera* L.	1	2

^1^ Jamun fruit (*Syzygium cumini)* is also referred to as “black plum”; ^2^ white sapote (*Casimiroa edulis)* is also referred to as “Mexican apple”.

**Table 4 nutrients-13-02417-t004:** Description of the different cluster groups and number of households within each group.

	Cluster Group	Description	Number of Households within the Group
1	The barrierless	Those who were able to implement all recommendations agreed upon without mentioning any difficulties.	10
2	The overcomers	Those who were able to implement all recommendations agreed upon despite mentioning difficulties.	14
3	The limited in fruit consumption and /or planting fruit trees	Those who were able to implement all recommendations agreed upon despite mentioning difficulties—except serving fruits on a daily basis and/or planting fruit trees.	18
4	The struggling	Those who were not able to implement all recommendations agreed upon.	6

**Table 5 nutrients-13-02417-t005:** Mean numbers of different types of fruit trees in the homestead per cluster group.

	1. The Barrierless	2. The Overcomers	3. The Limited in Fruit Consumption and/or Planting Fruit Trees	4. The Struggling
Number of fruit trees				
Min	1	1	1	1
Max	9	10	6	4
Mean ± SD	6.9 ± 2.4 ^a^	5.7 ± 2.2 ^a^	3.3 ± 1.6 ^b^	2.5 ± 1.0 ^b^
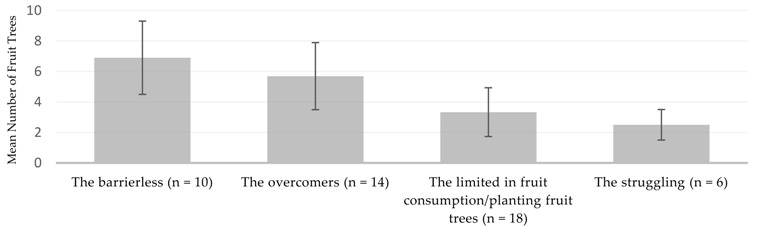				

a, b significant differences between the groups (*p* ≤ 0.05), a, a or b, b no significant differences.

## Data Availability

The raw data supporting the conclusions of this article will be made available by the authors, without undue reservation. The data are not publicly available due to privacy restrictions of the farm families.

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
