# Peer review of "Determinants of Children’s Fruit Intake in Teso South Sub-County, Kenya—A Multi-Phase Mixed Methods Study among Households with Children 0–8 Years of Age"

_nutrients, 2021, doi:10.3390/nu13072417_

Round 1

Reviewer 1 Report

This is a well written manuscript which investigated the factors affecting fruit intake of young children in Teso, Kenya. It provides insight and valuable information that could be implemented in future public health initiatives aiming to increase children’s fruit intake in this region.

In my view, the following points should be addressed.

Abstract:

On line 18, it is stated that 50 farm households were included in the study, however according to Figure 3 (page 7), 53 households were initially recruited of which 48 included in the analysis. Please clarify which number is correct.

Introduction:

The last two paragraphs (lines 83-93) should be part of the methods section.

The research ethics committee reviewing this study, is currently mentioned twice (lines 91 and 102). I suggest to merge these two points.

Methods:

You included families with children between 0-8 yrs old. Could you please explain what was the rationale behind this, especially since dietary practices for babies, toddlers and young children can vary a lot.

Figure 2: the figure legend and title are not embedded with the figure; they appear later on in the text (lines 135-139). Please format accordingly.

Table 1, last cell: ‘Interact with the child during mealtimes and actively and lovingly encourage it to eat’. Rephrase ‘it’ to him/ her. You cannot refer to a child as it..

Results:

Line 277: replace comma after ‘only’ with a full stop.

Table 3: add legend to include what superscripts 1 and 2 mean.

Table 5: on the legend you denote that * stands for p<0.05, but it does not appear anywhere on the data of the table. Please revise.

Table 5: a and b denote differences between groups, but the way it is phrased now is not clear to me between which groups. Please revise.

Discussion:

If I am not wrong, western Kenya has a wet and a dry season. Could you please explain how this could have affected fruit availability at the markets at the time of assessment? Can you explain if seasonality could also have affected the ability to plant a fruit tree, based on when TIPs were run?

Author Response

Dear reviewer, thank you very much for your comments which we used to improve our manuscript. Please see the attachment for our point-by-point response. Thank you.

Reviewer 2 Report

 Manuscript contains important data albeit the number of participants is too low. Authors should give explanation in Material and methods section why only 50 household were involved in study.  There is difficult to state that results are representative. even if the some criterions were described. It is still  not enough.

Please also add the following information in discussion section how long is going to take to obtain fruits from trees after starting horticulture. How it will affect the nutritional pattern of subpopulation and citizens from other parts of Kenya.

In opinion of reviewer before design of this study, authors should start with the food frequency questionnaire and nutritional knowledge questionnaire on bigger subpopulation from the same part of Kenya. It would give more information about the actual frequency of consumption of selected  group of foods and nutritional knowledge.

Author Response

Dear reviewer, thank you for your very valuable comments which helped us to improve our manuscript. Please find attached our point-by-point response for your information. Thank you.
